# Attribute-Driven or Green-Driven: The Impact of Subjective and Objective Knowledge on Sustainable Tea Consumption

**DOI:** 10.3390/foods12010152

**Published:** 2022-12-28

**Authors:** Manhua Zheng, Decong Tang, Anxin Xu

**Affiliations:** College of Economics and Management, Fujian Agriculture and Forestry University, Fuzhou 350002, China

**Keywords:** attribute knowledge, green knowledge, perceived product quality trust, purchase intention

## Abstract

The market for green agricultural products has tremendous growth potential as the pressure on resources and the environment increases and the safety of agricultural products is garnering attention. The demand for green food (tea) is also rising as tea is among the top three beverages consumed worldwide. The study attempts to propose a model of the relationship between green food (tea) customers’ product knowledge, perceived product quality, trust, purchase intention, and purchase behaviour. In addition, we will provide an analysis of the role played by age, education, income, gender, etc. The study included 700 questionnaires on green food (tea) consumers that were collected through the Credemo questionnaire platform, and data analysis was carried out using the SmartPLS software to assess the model of product knowledge on green food (tea) consumption behaviour. The findings demonstrate that, concerning differences in age, education, income, and gender; product knowledge—including attribute knowledge and green knowledge—has a positive impact on perceived product quality and trust; perceived product quality has a positive impact on trust; perceived product quality and trust have a positive impact on purchase intention; and purchase intention has a positive impact on purchase behavior.

## 1. Introduction

Global environmental problems (climate warming, glacier melting, soil and water pollution, etc.) frequently occur [1], triggering governments, companies, and consumers to pay attention to ecological and environmental protection [2]. Local governments have prioritized pollution control and environmental management, leading the industry toward a green transition and igniting a wave of enthusiasm for ecological environment protection [3]. A kind of environmental behavior that lessens environmental harm is purchasing environmentally friendly products [4]. Companies promote green food through media advertising to encourage consumers to purchase it, as it is a category of environmentally friendly goods that helps to safeguard the environment [2,5]. Currently, the tea industry faces issues with quality and safety, surface source pollution, and a lack of technology among tea farmers. Green tea (the green in this study refers to the sustainable characteristics of the food, not the color or variety of the food.) is a type of green food and is a safe and healthy, eco-friendly, green, and pollution-free product [6,7,8]. The tea industry’s sustainable and healthy growth depends on green food-tea development. The demand for green tea consumption is increasing as consumer consumption levels rise, government sustainability concepts mature, and consumer knowledge grows [6,9,10].

Consumers who lack knowledge of environmental issues are less likely to act in an environmentally friendly manner because knowledge is the biggest predictor of environmentally friendly behavior [11,12]. Product knowledge may be learned by consumers, and a lack of product knowledge can lead to gaps in attitudes and behaviors [13], and related research implies that product knowledge is an antecedent variable of purchase intention or purchase behavior [14,15]. However, there is considerable debate; on the one hand, some researchers contend that product knowledge has no direct effect on purchase intention or behavior [16,17]. On the other hand, several researchers have found that consumers’ intentions to make purchases or their actual behavior are not significantly influenced by their knowledge of a product [13,16,18], additionally, some studies [19,20] have come to the conclusion that knowledge is insufficient to motivate environmental behavior. There is debate on the connection between product knowledge and purchase intentions or behavior, presumably for three reasons.

First, it can be because the studies mentioned above did not focus on the same subjects (fabric softener, toilet paper, clothing, household appliances, lamps, batteries, etc.). Second, the above scholars do not define and classify green product knowledge in the same way, either as product knowledge, green firm knowledge, label knowledge and environmental knowledge, mostly as a higher-order unidimensional variable [21,22,23,24,25,26], and environmental knowledge refers to consumers’ perceptions related to climate change and the environmental impacts of production and consumption [27], which is an environmental perception of nature as a whole; and labelling knowledge, which refers to consumers’ ability to recognise labelling knowledge and understand the meaning of labels [28]. There is a gap between previous research on product knowledge and green food knowledge. This is because green food knowledge targets the food rather than the overall general environment, and green food contains not only green labels but also green attribute knowledge of overall production, transport and storage, as well as consumers’ subjective knowledge of how the overall process of green food production, transport and storage affects the environment, so it is necessary for this study to subdivide green food knowledge into two dimensions, namely attribute knowledge (objective) and green knowledge (subjective). Third, the intermediate variables between product knowledge and purchase intention or purchase behaviour are not the same (environmental attitudes, perceived value, perceived risk, perceived effectiveness, trust, environmental emotions), and there are fewer in-depth studies on purchase intention and behaviour from the respective sub-dimensions of product knowledge [15,20,26,29,30]. There are differences between green products and green food tea, with green products emphasizing the protection and improvement of the environment more [31] and green food tea having unique qualities, such as food safety attributes that offer consumers higher nutritional value to maintain their health [2,32]. A study based on the perspective of green food tea is, therefore, necessary. Taking care of this variety of difficulties can improve green consumption and therefore help to solve environmental issues.

There are two primary categories of a recent study on consumers’ purchase intentions of green products. First, from the standpoint of internal variables, researchers have found that consumers’ purchase intentions of green products are significantly influenced by attitudes, perceived value, perceived risk, perceived quality, perceived behavioural control, perceived effectiveness, and environmental concerns [14,33,34,35]. Second, from the standpoint of extraneous variables, researchers have found that brands, consumer reviews, and claims made in green advertising have a considerable positive influence on consumers’ intention to make green purchases [2,36,37]. Thus, it is clear that present researchers are less likely to examine consumer intention and behaviours related to the consumption of green foods (tea) from the standpoint of internal factors, such as product knowledge.

The study makes an effort to elaborate on the following research-related issues. Firstly, constructing a model of how product knowledge (attribute knowledge and green knowledge) affects consumers’ purchase intention and behaviour around the consumption of green food-tea. Secondly, expanding the areas in which consumer product knowledge theory can be applied. The study used green tea as its research object, and 700 green food consumer surveys were gathered to build a structural equation model with numerous clusters and to examine the model of consumer product knowledge on green tea purchase behaviour. This will serve as a guide for increasing green consumption and lowering the ecological and environmental pollution.

## 2. Research Hypotheses and Theoretical Framework

### 2.1. Research Hypotheses

#### 2.1.1. Attribute Knowledge and Perceived Product Quality

The word “attribute knowledge” describes pertinent, precise knowledge about the characteristics, lingo, frameworks, standards, etc., of green foods [38]. According to the external cue theory, consumers can form opinions about a product’s quality based on both internal and outward cues [39]. Consumers rely strongly on outward cues in a lemon market with asymmetric knowledge, while interior cues have little impact on product quality. When consumers buy green food, the green food certification label serves as an external cue with a signalling function that can provide some information about the food’s quality. This reduces information asymmetry in the market by providing consumers with valuable reference material on the product’s quality [40].

Consumers have more faith in independent third-party certification organizations than they have in the product’s manufacturers [41]. This is because third-party certifying bodies contribute to lowering the consumer risk since consumers believe they can establish and uphold just and equal standards, regulations, procedures, etc., in the exchange of goods [42]. Consumers find it challenging to spend more time and effort researching the safety of green foods, their nutritional value, the presence of pesticide residues, and other factors when food safety incidents are frequent and raise consumer doubts about the quality of food. The food certification label is a cheap way to screen products. Consumers’ views and trust in the quality of the food they are purchasing will improve if food is authorized by a third party to certify a high-quality product [43]. Studies have revealed that consumers’ views of product standards are influenced by the information they obtain from businesses or firms [44].

The hypothesis that is generated from the analysis above is as follows.

**Hypothesis 1 (H1)**.*Consumer attribute knowledge significantly improves the perceived product quality of green tea*.

#### 2.1.2. Attribute Knowledge and Trust

Compared to regular food, green food has qualities such as being safe, healthy, and environmentally friendly [2]. Additionally, consumers can only recognize green food at the point of purchase based on the green food label approved by a third-party certification body and the packaging promotion of the particular food [45]. This gives green food the quality of trust [46]. Third-party certification bodies reduce the opportunistic behaviour of businesses by setting rules, standards, and procedures (e.g., QS certification, HACCP certification, ISO series certification, green food certification, organic food certification, pollution-free agricultural products certification, GMO certification, etc.), ensure that qualified and compliant food products enter the market, and provide a comparatively safe and secure market for consumers to trade. As a result, via qualified, fair, and equitable transactions, consumers’ views of the qualities of green food are improved, and trust in green food firms, green food standards, and green food labelling is increased [41]. When food label information is relatively scarce, consumers will be more cautious and less trusting of the product; when label information is abundant, consumers will be more knowledgeable about the product and have higher levels of objective understanding about green food [47].

The hypothesis that is generated from the analysis above is as follows.

**Hypothesis 2 (H2)**.*Consumer attribute knowledge significantly improves trust in green tea*.

#### 2.1.3. Green Knowledge and Perceived Product Quality

The term “green knowledge” refers to consumers’ subjective understanding of the environmental and food safety benefits of “green” foods [38]. It is also possible to define “green knowledge” as consumers’ understanding of the impacts of reducing environmental pollution by purchasing “green” items. According to the quality signalling theory, consumers evaluate a product’s perceived quality using a range of information, and since green knowledge is embedded within the product, it affects how consumers view the product’s quality.

Consumers with green knowledge are often more concerned about environmental and ecological issues, and they frequently relate products with ethics [48,49]. Consumers may believe that green food is of higher quality than ordinary food since it is healthier, more ecologically friendly, and safer due to its ability to mitigate environmental problems [10,50].

The hypothesis that is generated from the analysis above is as follows.

**Hypothesis 3 (H3)**.*Consumer green knowledge significantly improves the perceived product quality of green tea*.

#### 2.1.4. Green Knowledge and Trust

Related research has demonstrated that as the market for green products grows and green information spreads broadly, consumers begin to feel more trust in green items [14]. Particularly, consumers are more likely to trust green food if they are more aware of it [22]. Consumers’ lack of understanding of the information on energy conservation and emission reduction provided by green products and their consequent mistrust of the information is one of the factors contributing to their difficulties in making purchasing decisions [51]. Consumers’ subjective assessments of the environmental protection efforts made by green food producers and the high level of food safety will strengthen their confidence in the capability of green food to lessen environmental pollution, supply nourishment, and sustain health.

The hypothesis that is generated from the analysis above is as follows.

**Hypothesis 4 (H4)**.*Consumer green knowledge significantly increases trust in green tea*.

#### 2.1.5. Perceived Product Quality and Trust

As consumers base their purchasing decisions on incomplete and asymmetrical information, the perceived product quality is a signal from the company to the consumer based on which the consumer develops a sense of trust [52]. Consumers lose faith in a product when a company makes green claims that do not match how they think the product is made [53,54]. Consumers will have a greater perception of trust in a product or brand when they believe it to be of a higher quality [55,56]. Related studies have shown that perceived quality positively influences consumers’ trust [25,57,58].

The hypothesis that is generated from the analysis above is as follows.

**Hypothesis 5 (H5)**.*Consumer-perceived product quality significantly increases trust in green tea*.

#### 2.1.6. Perceived Product Quality and Purchase Intention

By affecting the perceived advantages, which are higher than the costs when the perceived quality is higher, the perceived quality of the product or service can improve the consumer’s purchase intention [59]. When making food purchases, consumers should consider the product’s quality [60,61]. When consumers see green food as having higher quality than regular food (natural and unadulterated, free of dangerous elements, chemical fertilizers, etc.), they are more likely to purchase it [62,63]. Contrarily, agricultural products with lower perceived product quality (broken packing, deformed appearance, flaws, etc.) considerably decrease consumers’ intention to make a purchase [64].

The hypothesis that is generated from the analysis above is as follows.

**Hypothesis 6 (H6)**.*Consumer-perceived product quality significantly increases the purchase intention of green tea*.

#### 2.1.7. Trust and Purchase Intention

Consumers’ trust in food firms’ products regarding food safety, health, and environmental protection is referred to as trust. Due to the unique characteristics of green foods and the information asymmetry in the market, consumers can only evaluate the quality of a product through personal experience and product expertise, making the intake of green food risky [41]. When consumers believe there is a significant risk involved in the consumption process, their faith in food producers, food certification organizations, marketing, green labelling, etc., helps to lower the perceived risk and thus increases their desire to acquire green food [65].

According to trust theory, a high level of trust is necessary for the establishment of a long-term, stable relationship with consumers and the creation of value in their transactional relationship. This helps consumers form a good corporate image and increases their likelihood of making a purchase [66]. Related research has demonstrated that green trust has a considerable, favourable impact on green purchase intention [33,67].

The hypothesis that is generated from the analysis above is as follows.

**Hypothesis 7 (H7)**.*Consumer trust significantly increases the purchase intention of green tea*.

#### 2.1.8. Purchase Intention and Purchase Behaviour

According to the Theories of Planned Behaviour and Rational Behaviour, an individual’s behaviour can be inferred from behaviour intention. Behaviour intention is a measure of behaviour [17]. According to a related study, green purchase behaviour is positively influenced by green purchase intention [34,68,69,70]. The hypothesis that is generated from the analysis above is as follows.

**Hypothesis 8 (H8)**.*Consumer purchase intention significantly increases the purchase behaviour of green tea*.

### 2.2. Theoretical Framework

The SOR (Stimulus-Organism-Response) theoretical model has been used extensively in the field of consumer behaviour [71]. SOR theory suggests that external stimuli (products, brands, marketing, etc.) can influence consumers’ intentions and behaviour by affecting their emotions and perceptions (perceived quality, perceived value, perceived risk, etc.) [56,72,73]. In this study, attribute knowledge and green knowledge of green tea are used as external Stimulus variables, perceived product quality and trust as Organism variables, and purchase intention and purchase behaviour as response variables (Response). Constructing the theoretical model framework of this study based on the SOR model Integrating the above analysis, the theoretical model of this study is derived, as shown in Figure 1.

## 3. Methods

### 3.1. Participants and Procedure

The sample size of the study was calculated using Gpower3.1 and with an effect size f = 0.02, significance level = 0.05, test validity power = 0.80, and tested predictors = 5, the sample size was 647, therefore the study set a sample size of 700, which is sufficient to justify the adequacy of the study sample [74].

The study was based on a questionnaire collected by Credamo, an internationally recognized questionnaire collection platform, and an additional RMB 595 was spent to set the completion conditions to ensure the quality of the returned sample. Firstly, a credit score of 80 or above was set, as the higher the credit score of the sample, the higher the quality of the completed questionnaire. Secondly, we set the sample’s historical adoption rate to be greater than or equal to 80 to avoid the low quality of the questionnaire due to the unfamiliarity of the process. Thirdly, a sliding puzzle was set for intelligent human verification before answering, which helped to improve data quality and security. Fourthly, only one participant within 5 km was set to avoid sample homogeneity due to the bunching of participants.

The questionnaire was randomly distributed to all provinces in China in September 2022, and 700 questionnaires were returned, with a participant reward amount of RMB 3 per questionnaire. Two screening questions were set, the automatic screening question “20 + 40 = ?” and “How often did you buy green tea in one year (repeated question)”. During data cleaning, samples with inconsistent responses were removed to obtain the final sample for this study to ensure the reliability of the sample.

### 3.2. Measures

The overall questionnaire is divided into three main blocks. The first block is a brief graphic introduction to green foods. The second block contains questions related to the study variables (all borrowed from previously established scales and adapted to the actual situation of green tea, the measurement table is shown in Table 1), measured on a 7-point Likert scale. The third block contains questions on the demographic characteristics of the participants (gender, age, education, income, occupation, and household registration).

## 4. Results

### 4.1. Demographic Profile

SPSS software was applied to conduct a descriptive statistical analysis of the sample and the results were shown in Table 2. Based on the demographic characteristics, it can be seen that the majority of participants were female (63.3%), which is in line with the perception that the majority of people responsible for purchasing food in households are female [46], similar to previous related studies on green products [32], which may be due to the fact that the main population of green food purchasers in Chinese households is female [80]. The participants were mostly urban and preferred national and local green food, most likely because green food stores are mainly located in cities and Chinese consumers prefer local food, and imported food does not appear to be as fresh as local food due to long distance transportation [81]. The age is concentrated in the age group of 23–40 years old (70.3%), mostly young people, which may be due to the fact that this group has a higher spending power and is the mainstream group of consumption, similar to related studies [24,82]. The education level was high, with 70.1% of the respondents having a bachelor’s degree. The participants were mostly urban and preferred national and local green food, most likely because green food stores are mainly located in cities and Chinese consumers prefer local food, and imported food does not appear to be as fresh as local food due to long distance transportation [81] Monthly income was mainly concentrated in the range of RMB 4000–8000. Most of the participants were urban residents and favoured national and local green food.The overall sample characteristics of the study were similar to those of previous related studies on green foods in China [2,80], which are representative.

### 4.2. Reliability and Validity Analysis

To test the structural equation model, the study used SmartPLS3. Compared to the covariance-based structural equation modelling data analysis tool, this software considers the feasibility of all path coefficients, has no requirement for the number of question terms of the variables in the model, and is able to obtain more robust estimation results.. First, the scale’s reliability was evaluated. Referring to previous studies related to SmartPLS, this study applied three measures of Cronbach’s Alpha, Rho_A, and Composite Reliability to assess reliability [83]. The results are provided in Table 3, and the figures are higher than the critical values of 0.6, 0.7, and 0.8, respectively, suggesting that the scale has good reliability [2].

The scale’s validity was then evaluated. Validity was evaluated using the average variance extracted value (AVE), external factor loadings, FORNELL, and heterotrait-monotrait (HTMT) ratio. The external factor loadings were all more than 0.7, as shown in Table 4, indicating that all items on the scale had favourable external loadings [84]. All constructs had AVEs of more than 0.5, as shown in Table 3, indicating a variance of more than 50% and good convergent validity [85].

As can be seen in Table 5, according to the conclusion of previous studies, good discriminant validity is indicated when the square root of AVE of all the constructs is higher than the absolute value of Pearson correlation coefficient of each construct under that column [85], and the results of the present study results meet this condition. In Table 6, it is evident that the HTMT between the two constructs was less than 0.90, further supporting high discriminant validity [86] and providing circumstantial evidence that the data were not multicollinear [87].

### 4.3. Assessment of Structural Model

The findings of the model evaluation using a bootstrapping subsamples 5000 study are displayed in Table 7 and Table 8. According to the conclusion of previous studies, when the variance inflation factor (VIF) is less than the critical value of 5, it indicates that there is no covariance problem in the data as well as common method bias problem [88], and the results of this study meet the condition.

All constructs are higher than the crucial value of 0.25, as determined by $R^{2}$, indicating that the model has a good predictive capacity [89]. All constructs larger than 0.15 indicate medium forecasting ability and are greater than $Q^{2}$, indicative of out-of-sample predictive association [90]. The Standardized root means square residual (SRMR) is an absolute measure of model fit with SRMR = 0.063 < 0.080, indicating good model fit superiority [91].

The study employed PLS-SEM to assess the model’s hypotheses, and the data’s findings are displayed in Table 9. According to previous studies, it is known that the model passes the significance test when the test result t-value is greater than 1.96 or the *p*-value is less than 0.05 [86,92], and all hypotheses in this study passed the significance test. Attribute knowledge had a significant positive effect on perceived product quality (β = 0.306, t = 6.408, *p* < 0.000 and did not cross 0 at the 5–95% confidence interval), and H1 was tested. Attribute knowledge had a significant positive effect on trust (β = 0.116, t = 3.091, *p* = 0.001 and did not cross 0 at 5–95% confidence intervals), and H2 was tested. Green knowledge had a significant positive effect on perceived product quality (β = 0.435, t = 10.795, *p* < 0.000 and did not cross 0 at 5–95% confidence interval), and H3 was tested. Green knowledge had a significant positive effect on trust (β = 0.224, t = 5.746, *p* < 0.000 and did not cross 0 at 5–95% confidence interval), and H4 was tested. Perceived product quality had a significant positive effect on trust (β = 0.505, t = 13.336, *p* < 0.000 and did not cross 0 at 5–95% confidence interval), and H5 was tested. Perceived product quality had a significant positive effect on purchase intention (β = 0.517, t = 11.065, *p* < 0.000 and did not cross 0 at 5–95% confidence interval), and H6 was tested. Trust had a significant positive effect on purchase intention (β = 0.288, t = 5.663, *p* < 0.000 and did not cross 0 at 5–95% confidence interval), and H7 was tested. Purchase intention had a significant positive effect on purchase behaviour (β = 0.514, t = 18.953, *p* < 0.000 and did not cross 0 at 5–95% confidence interval), and H8 was tested.

### 4.4. Multiple Group Analysis

The outcomes of the multi-group analysis are displayed in Table 10. The results of the multi-group analysis, which divided the subjects’ educational levels into low education (undergraduate and below) and high education (undergraduate and above), show that attributing knowledge and green knowledge to trust had no significant influence on subjects with lower educational levels.

The results of the multi-group analysis, which split the respondents’ ages into middle-aged and older (30 years and older) and young (30 years and below) groups, showed that the influence of attribute knowledge on trust was not significant for older individuals. Multiple analyses of gender revealed that the relationship between attribute knowledge and trust was not significant for male individuals. From the overall results, those with relevant knowledge background, youth, and females, have higher awareness of green food labels, and label awareness further influences consumers’ label attitude evaluation (trust). This may be because groups with better knowledge of attributes are more willing to trust green food labels and make positive evaluations, building positive beliefs based on the knowledge gained [93].

## 5. Discussion

Previous studies have shown that knowledge is an important variable in predicting consumer pro-environmental behaviour [11]. However, existing studies are controversial about how knowledge affects consumer behaviour and the segmentation of product knowledge varies [22,23,24,26,30]. Therefore, it is necessary to develop a study of consumer behaviour by segmenting product knowledge from a green food perspective.

First, it deepens the research related to green product knowledge. Previous studies on green product knowledge have mainly been conducted from the perspectives of environmental knowledge, overall knowledge, subjective knowledge, brand knowledge, cost knowledge, and quality knowledge [15,22,24,26,51,94,95], and some of the studies have been conducted on green knowledge, but in this study green knowledge was considered as consumer knowledge related to information about product attributes [95]. However, according to previous related studies, product knowledge can be divided into subjective and objective knowledge [96], therefore, this study divides green tea product knowledge into attribute knowledge (objective knowledge) and green knowledge (subjective knowledge), which helps to study the research related to product knowledge from the perspective of lower-order segmentation.

Second, it is necessary to improve the mechanism of the influence of product knowledge on green tea consumption behaviour. There are previous studies related to product knowledge on purchase intention, purchase intention on purchase behaviour, trust on purchase intention, and knowledge on trust [14,22,25,26,67], but the findings of these studies are inconsistent, with either a direct effect or the presence of other variables between the two quantitative variables playing an indirect effect. There are fewer studies related to the relationship between the other variables knowledge on perceived product quality, perceived product quality on trust, and perceived product quality on purchase intention between the two. Therefore, this study constructs a new model framework to study the influence of product knowledge on consumption behaviour that is meaningful and enriches the related research on green tea consumption intention and behaviour.

Third, the effect of green knowledge on perceived product quality was greater than that of attribute knowledge. This finding is similar to the findings of previous studies on environmental knowledge and label knowledge on attitudes [22], but the difference is that the green food knowledge in this study targets food products rather than people’s perceptions of the general environment, and green food attribute knowledge contains not only label knowledge but also perceptions of overall production, transportation, and storage. Consumers’ product knowledge had a significant positive effect on trust, with green knowledge having a greater effect than attribute knowledge. This finding is similar to previous studies [28], but differs in that the study only concluded that label knowledge can have an effect on trust, while environmental knowledge has no effect on trust.

Fourth, perceived product quality has a significant positive effect on trust. This finding is similar to that of previous studies on millennial samples (information quality positively affects brand trust), but differs in three ways. First, this study sample is more generalizable to the entire social sample. Second, that study was on brands, while this study is on food products, which extends the scope of the study. Finally, there is a gap between information quality and perceived product quality; information quality refers to the validity, accuracy, and timeliness of information, while perceived product quality focuses on consumers’ perceptions of the product [25].

Fifth, perceived product quality has a significant positive effect on purchase intention. This finding is similar to the findings of previous studies on suboptimal agricultural products, but with the following differences. The study started the research on food products from a negative perspective, i.e., lower quality food (damaged packaging, cosmetic defects, and nearing the end date), while this study started the research on food products from a positive perspective, i.e., green tea (environmentally friendly and non-polluting, nutritious, and guaranteeing food safety), which helped to broaden the scope of application of the study [64].

Sixth, trust has a significant positive effect on purchase intention. This finding is similar to the findings of a previous study (green trust positively affects green purchase intention) for electronic products in Taiwan [33]. This suggests that the findings can be applied to the food sector, due to the fact that consumer trust is an essential factor for long-term behaviour [97], and that consumers are more likely to purchase if they trust the seller’s experience (environmental attributes of the product and health attributes of the food). If firms exaggerate the environmental performance of their products, consumers are less likely to trust the firm [98] and thus less likely to buy. Similarly, if a company exaggerates the food safety and health attributes of green tea, consumers will have a feeling of false propaganda towards the company and thus become suspicious of green tea, and thus are reluctant to buy green tea.

Seventh, purchase intention has a significant positive effect on purchase behaviour. This finding is similar to previous studies on overall green products (green purchase intention positively affects green purchase behaviour) [34], but with the following differences. On the one hand, the study was conducted for the overall green product domain and the findings may not be representative of the findings for each sub-category of green products. On the other hand, the measurement scales of green purchase intention and green purchase behaviour in the overall green product study are different from the measurement scales of the variables in this study, which suggests that the findings are consistent across quantitative variables, again confirming the generalizability of the study findings [2].

## 6. Limitations and Future Research

### 6.1. Limitations

First, the data in this study were collected at a single point in time, and there may be seasonal influences that do not accurately predict changes in green food consumer attitudes and behaviours throughout the year. Second, the data in this study were collected in China, and there may be influences from the degree of economic development, consumption habits, consumer attitudes, and national cultures of different countries. Without the influence of these boundary conditions, the findings may not accurately predict the behaviour of green food consumers in every country in the world.

### 6.2. Future Research

First, future research can be launched to collect data at different time points and conduct in-depth tracking studies. Data from multiple periods can be used to analyse the dynamics of consumer purchasing behaviour. It is also possible to use experimental methods for analysis, to control the main research variables, and to find the impact of variables such as time point and season. Second, the number of questionnaires distributed and the scope of distribution should be broadened so that the sample can cover all types of countries more comprehensively and make the sample more representative. Future research can distribute questionnaires to different countries, add boundary variables such as economic development level, consumption habits, culture, etc., involve more consumer groups, and conduct cross-regional comparative analysis.

## 7. Conclusions

Based on the SOR model, this study investigates the consumption behaviour of green tea by using green knowledge and attribute knowledge as external stimulus variables, perceived product quality and trust as mechanism variables, and purchase intention and behaviour as response variables, and draws the following conclusions. Consumers’ attribute knowledge and green knowledge have a significant positive effect on perceived product quality and trust, most likely because green knowledge, compared to attribute knowledge, is part of the quality perceptions of the product that consumers have already formed in their subjective consciousness about green tea, and thus green knowledge has a higher impact on the product quality of green tea [99]. Green knowledge is the consumers’ perception of subjective knowledge of green tea’s contribution to mitigating climate change, reducing resource consumption, and reducing environmental pollution, and there are changes in consumers’ internal attitudes in this perception process, and these are part of the functions and utilities of green tea, which make consumers more confident in making judgments about purchasing green tea [14]. In contrast, attribute knowledge is only the perception of green tea meaning, system, label, etc., which belongs to objective perception, but can be considered as an external stimulus. Therefore, changes in internal attitudes have a higher impact on consumer trust compared to external stimuli [22]. Perceived product quality has a significant positive effect on trust, most likely because consumers are in a market with information asymmetry and consumer trust is dependent on perceived product, brand, and service quality [52,57,100]. Perceived product quality has a significant positive effect on purchase intention, suggesting that perceived quality is an important factor for firms to maintain long-term relationships with consumers [101], providing a reason for consumers to purchase green tea and differentiating it from other teas [102]. Trust has a significant positive effect on purchase intention, and purchase intention has a significant positive effect on purchase behaviour. On this basis, the following recommendations are made.

Firstly, when designing the packaging of tea products, green tea information should be displayed to the greatest extent on the outer packaging so that consumers can quickly access green tea-related knowledge. For example, the green food certification can be designed in a prominent position on the outer packaging.

Secondly, when designing green tea publicity manuals, one should add various standards related to green tea from production, origin, storage, transportation, use of chemical fertilizers and pesticides, etc., in the manuals, and compare them with ordinary tea to cultivate consumers’ subjective green knowledge of green tea from objective attribute knowledge.

Thirdly, one should enhance the advertising and publicity of green tea. Green tea advertisements can be placed on TV, Taobao, JD, PDD, Tik Tok, Quick hand, WeChat, QQ, Weibo, Red, Bilibili, buses, and taxis to increase product publicity and enhance consumers’ product knowledge awareness, product quality perception, and product trust in green tea.

Fourthly, enhancing product knowledge training for green tea sales staff by enterprises is conducive to introducing consumers to the attributes of green tea such as health and pollution-free, food safety, and environmental protection before sales, thereby enhancing consumers’ product knowledge.

Fifthly, through technical innovation and service enhancement, consumer trust is enhanced. On the one hand, through technological innovation, the quality of green tea (taste, nutrition, and appearance) will be improved, and corporate self-inspection and market inspection of green tea product quality will be carried out to prevent poor-quality green tea from using green food certification to deceive consumers, thereby enhancing consumers’ trust in the product. On the other hand, one should build green tea communities to provide consumers with perfect pre-sales and after-sales services, and actively communicate with consumers, thereby enhancing the consumers’ perception of product quality and consumers’ trust in the products.

Sixth, one should target different consumer segments and conduct precision marketing. Target marketing should be aimed at consumers of different ages, education, income, and gender.

## Figures and Tables

**Figure 1 foods-12-00152-f001:**
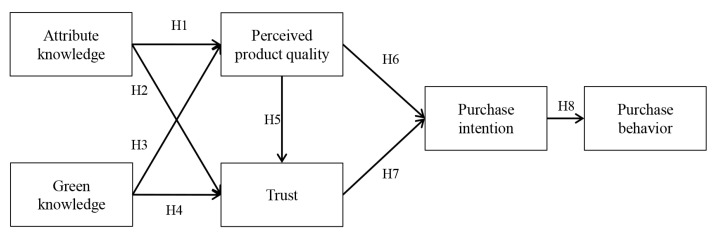
Model of the influence of consumer product knowledge on green tea consumption behaviour.

**Table 1 foods-12-00152-t001:** Measurement.

Variables	Items	Sources
Attribute knowledge	I have heard the term green tea	Alba and Hutchinson [75]
	I understand the meaning of green tea	
	I understand the system, standards and other information related to green tea	
Green knowledge	Buying green tea can mitigate climate change	Mohd Suki [24]
	Buying green tea can reduce the current level of environmental pollution	
	Buying green tea can reduce the consumption of natural resources	
	Buying green tea means I am doing the right thing	
Perceived product quality	I think green tea has higher nutritional value	Yoo et al. [76]
	I think the quality and safety of green tea is guaranteed	
	I think green tea is good for the environment and in line with the concept of sustainable development	
	Green tea meets my needs	
Trust	Green certified tea is safe, non-polluting, and environmentally friendly	Zhao et al. [77]
	I think the green tea certification organization is more credible	
	I believe that the green tea certification process is fair and credible	
	I believe that the information shown by green tea is true	
Purchase intention	I am willing to collect and learn about green tea	Kozup et al. [78]
	I consider buying green tea	
	I would recommend green tea to others	
	I am willing to pay more for green tea	
	I am willing to buy green tea if I have enough time, energy and money	
Purchase Behaviour	Frequency of purchasing green tea in a year	Li et al. [79]

**Table 2 foods-12-00152-t002:** Descriptive statistics (N = 700).

	Items	Frequency	Proportion
Gender	Female	443	63.3%
	Male	257	36.7%
Age (in years)	22 and below	143	20.4%
	23–30	299	42.7%
	31–40	193	27.6%
	41–50	44	6.3%
	51–60	18	2.6%
	61 and above	3	0.4%
Education	High School and below	42	6.0%
	College	84	12.0%
	Undergraduate	491	70.1%
	Postgraduate and above	83	11.9%
Job	Public institution	78	11.1%
	Civil servants	23	3.3%
	Other	17	2.4%
	State-owned enterprises	139	19.9%
	Foreign-funded enterprises	25	3.6%
	Students	150	21.4%
	Private enterprises	251	35.9%
	Freelancers	17	2.4%
Monthly income (RMB/Yuan)	Below 4000	187	26.7%
	4000–8000	237	33.9%
	8000–12,000	184	26.3%
	12,000 and above	92	13.1%
Household Registration	Rural	270	38.6%
	Urban	430	61.4%
Place of origin1	Field	43	6.1%
	Local	657	93.9%
Place of origin2	Foreign	18	2.6%
	National	682	97.4%

**Table 3 foods-12-00152-t003:** Reliability and validity analysis.

Constructs	Cronbach’s Alpha	Rho_A	Composite Reliability	AVE
Attribute knowledge	0.812	0.817	0.889	0.728
Green knowledge	0.847	0.858	0.896	0.684
Perceived product quality	0.743	0.750	0.838	0.564
Trust	0.825	0.825	0.884	0.656
Purchase intention	0.858	0.859	0.898	0.637
Purchase behavior	1.000	1.000	1.000	1.000

**Table 4 foods-12-00152-t004:** Factor loadings and VIF.

Constructs	Indicator	Indicator Reliability	VIF
Attribute Knowledge	AK1	0.809	1.575
	AK2	0.873	1.959
	AK3	0.876	2.011
Green knowledge	GK1	0.796	1.969
	GK2	0.857	2.277
	GK3	0.85	2.165
	GK4	0.803	1.555
Perceived product quality	PPQ1	0.733	1.431
	PPQ2	0.767	1.504
	PPQ3	0.71	1.391
	PPQ4	0.793	1.488
Trust	T1	0.795	1.657
	T2	0.784	1.635
	T3	0.826	1.853
	T4	0.834	1.897
Purchase intention	PI1	0.771	1.714
	PI2	0.795	1.832
	PI3	0.784	1.83
	PI4	0.813	2.004
	PI5	0.827	2.011
Purchase behavior	PB	1	1

**Table 5 foods-12-00152-t005:** Discriminant validity (FORNELL).

	(1)	(2)	(3)	(4)	(5)	(6)
Attribute Knowledge (1)	0.853					
Green knowledge (2)	0.463	0.827				
Perceived product quality (3)	0.507	0.576	0.751			
Trust (4)	0.476	0.569	0.693	0.81		
Purchase intention (5)	0.582	0.599	0.717	0.646	0.798	
Purchase behavior (6)	0.535	0.393	0.414	0.374	0.514	1

Note: The square root of AVE is shown on the diagonal, while the absolute value of the Pearson correlation coefficient for each variable is shown below the diagonal.

**Table 6 foods-12-00152-t006:** Discriminant validity (HTMT).

	(1)	(2)	(3)	(4)	(5)
Green knowledge (1)	0.551				
Perceived product quality (2)	0.643	0.551			
Trust (3)	0.580	0.667	0.884		
Purchase intention (4)	0.696	0.689	0.888	0.767	
Purchase behavior (5)	0.593	0.424	0.472	0.411	0.555

**Table 7 foods-12-00152-t007:** Internal model VIF values.

	(1)	(2)	(3)	(4)
Attribute Knowledge (1)	1.272	1.429		
Green knowledge (2)	1.272	1.590		
Perceived product quality (3)		1.681	1.923	
Trust (4)			1.923	
Purchase intention (5)				1

**Table 8 foods-12-00152-t008:** Model results.

	$R^{2}$	$Q^{2}$	SRMR
Attribute Knowledge			0.063
Green knowledge			
Perceived product quality	0.405	0.219	
Trust	0.533	0.344	
Purchase intention	0.557	0.350	
Attribute Knowledge	0.264	0.261	

**Table 9 foods-12-00152-t009:** Hypothesis testing.

Hypotheses	β	LLCI	ULCI	Mean	SD	T	*p*	Status
H1: Attribute Knowledge -> Perceived Product Quality	0.306	0.226	0.384	0.306	0.048	6.408	0	Supported
H2: Attribute knowledge -> Trust	0.116	0.055	0.178	0.117	0.038	3.091	0.001	Supported
H3: Green knowledge -> Perceived product quality	0.435	0.37	0.502	0.436	0.04	10.795	0	Supported
H4: Green knowledge -> Trust	0.224	0.159	0.288	0.224	0.039	5.746	0	Supported
H5: Perceived product quality -> Trust	0.505	0.441	0.566	0.504	0.038	13.336	0	Supported
H6: Perceived product quality -> Purchase intention	0.517	0.437	0.591	0.516	0.047	11.065	0	Supported
H7: Trust -> Purchase intention	0.288	0.206	0.373	0.29	0.051	5.663	0	Supported
H8: Purchase intentions -> Purchase behavior	0.514	0.47	0.559	0.515	0.027	18.953	0	Supported

**Table 10 foods-12-00152-t010:** Multigroup analysis result.

Hypotheses	Education		Age		Gender	
	*p* (Low)	*p* (High)	*p* (Mid and Older)	*p* (Young)	*p* (Female)	*p* (Male)
H1: Attribute Knowledge -> Perceived Product Quality	0.000	0.000	0.000	0.000	0.000	0.000
H2: Attribute knowledge -> Trust	0.155	0.003	0.056	0.011	0.001	0.153
H3: Green knowledge -> Perceived product quality	0.000	0.000	0.000	0.000	0.000	0.000
H4: Green knowledge -> Trust	0.293	0.000	0.008	0.000	0.000	0.037
H5: Perceived product quality -> Trust	0.000	0.000	0.000	0.000	0.000	0.000
H6: Perceived product quality -> Purchase intention	0.000	0.000	0.000	0.000	0.000	0.000
H7: Trust -> Purchase intention	0.037	0.000	0.001	0.000	0.000	0.008
H8: Purchase intention -> Purchase behavior	0.000	0.000	0.000	0.000	0.000	0.000

## Data Availability

The data presented in this study are available on request from the corresponding author.

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
