# Peer review of "Attribute-Driven or Green-Driven: The Impact of Subjective and Objective Knowledge on Sustainable Tea Consumption"

_foods, 2022, doi:10.3390/foods12010152_

Round 1

Reviewer 1 Report

Thank you very much for giving me the opportunity to read the manuscript The impact of attribute knowledge (objective) and green knowledge (subjective) on green food(tea) consumption. The topic is interesting and relevant to FOODS readership. The paper contributes to consumer preferences' literature by seeking to demonstrate the existence of relationship between green food (tea) 4 customers’ product knowledge, perceived product quality, trust, purchase intention, and purchase behaviour.

The paper is well written and structured, answers hypothesis, and the contents are easy to follow. English spelling and grammar are good.

 1 Introduction

This section clearly focuses on the aim of the manuscript, it flows nicely. Authors should clarify at the beginning that green is not a colour or a variety but express a sustainability characteristic. Maybe title could help to clarify better this concept.

2. Research hypotheses

Despite a specific section of the literature is absent. Research hypotheses compensate this lacking and help to readers to clearly understand focus of study and objectives. Therefore, in my opinion, authors should add in this section a brief description of the conceptual frameworks in which their study falls.

3. Methods-

The data collection and analysis methods used in the research are clearly explained, in a way that other researchers could repeat it.

I only suggest to specify if sample is representative of the entire nation.

In addition is rather unclear the interpretation of following statement; the automatic screening question 230 "20+40+? (line 230).

Finally, I should summarize all subsections from 3.21. to 3.2.6 in a synthetic table

 4 Results and discussion

The results are presented in a comprehensive way and appropriate sequence and all data presented in tables and figures are easily understandable.

I have only a question:(pag 9 Line 292) - What does it mean “bootstrapping subsamples 5000 study are 292 displayed in Tables”?

Lastly, I suggest to remove the brackets and avoid word repeated twice (Knowledge and green) in the main title. I suggest to find a more attractive title.

Reviewer 2 Report

I have reviewed the manuscript titled “The impact of attribute knowledge (objective) and green knowledge (subjective) on green food (tea) consumption” thoroughly and it needs the revisions proposed below.

Introduction

1.      Environmental issues such as climate change, glacial melting, soil and water contamination, etc. are widespread [1], raising concern among governments, businesses, and consumers worldwide [2]. Please rewrite this sentence.

Methods

1.      The sample size of the study was calculated using Gpower software. Please provide complete detail of software like version etc. Also why you have selected this software?

2.      The questionnaire was randomly distributed to all provinces in China in September 228 2022. How you distributed the questionnaire throughout the China?

3.      Please provide sample questionnaire.

Results

1.      Based on the demographic characteristics, it was clear 265 that the majority of participants were female (63.3%). What is the reason? Support you reasoning with the help of available literature.

2.      Age was concentrated 268 in the age group of 23-40 years (70.3%), mostly young people. What is the reason? Support you reasoning with the help of available literature.

3.      Most of the participants were urban residents and favored national and local green food. What is the reason? Support you reasoning with the help of available literature.

4.      To test the structural equation model, the study used SmartPLS software. Please provide complete detail of software like version etc. Also why you have selected this software?

5.      The reliability was evaluated using Cronbach’s Alpha, Rho A, and Composite Reliability measures. Why you have used this?

6.      As can be seen in Table 4, excellent discriminant validity is shown by the square root of the AVE for each construct is higher than the absolute value of the Pearson correlation coefficient for each construct beneath the column. What is the reason? Support you reasoning with the help of available literature.

7.      The variance inflation factors (VIF) were all below the threshold value of 5, demonstrating that the data were free of covariance and typical technique bias problems. What is the reason? Support you reasoning with the help of available literature.

8.      Attribute knowledge had a significant positive effect on perceived product quality (β = 0.306, t = 6.408, p < 0.000 and did not cross 0 at the 5%-95% confidence interval) and H1 was tested. What is the reason? Support you reasoning with the help of available literature.

9.      Attribute knowledge had a significant positive 304 effect on trust (β = 0.116, t = 3.091, p = 0.001 and did not cross 0 at 5%-95% confidence intervals) and H2 was tested. What is the reason? Support you reasoning with the help of available literature.

10.  Green knowledge had a significant positive effect on perceived product quality (β = 0.435, t = 10.795, p < 0.000 and does not cross 0 at 5%-95% confidence interval) and H3 was tested. Green knowledge had a significant positive effect on trust (β = 0.224, t = 5.746, p < 0.000 and does not cross 0 at 5%-95% confidence interval) and H4 was tested. What is the reason? Support you reasoning with the help of available literature.

11.  Perceived product quality had a significant positive effect on trust (β = 0.505, t = 13.336, p < 0.000 and does not cross 0 at 5%-95% confidence interval) and H5 was tested. What is the reason? Support you reasoning with the help of available literature.

12.  Perceived product quality had a significant positive effect on purchase intention (β = 0.517, t = 11.065, p < 0.000 and does not cross 0 at 5%-95% confidence interval) and H6 was tested. What is the reason? Support you reasoning with the help of available literature.

13.  The results of the multi-group analysis, which split the respondents’ ages into middle aged and older (30 years and older) and young (30 years and below) groups, showed that the influence of attribute knowledge on trust was not significant for older individuals. What is the reason? Support you reasoning with the help of available literature.

Discussion

Discussion is weak. It should be revised by compare the present finding with the available literature.

Conclusion

1.      Write it separately from the discussion.

2.      Please add your result highlights to the conclusion.

3.      What are your future recommendations?

Limitations and future research

Write this part between discussion and conclusion.

Round 2

Reviewer 2 Report

I have re-evaluate the manuscript title “The impact of attribute knowledge (objective) and green knowledge (subjective) on green food (tea) consumption” thoroughly and found that authors have revised the manuscript according to the suggestions proposed.